# PanoptiChrome: A Modern In-browser Taint Analysis Framework

Anonymous

## ABSTRACT

Taint tracking in web browsers is a problem of profound interest because it allows developers to accurately understand the flow of sensitive data across Javascript (JS) functions. Given that modern websites load JS functions from either the web server or from other third-party sites, this problem has acquired a much more complex and pernicious dimension. Sadly, for the latest version of the Chromium browser (used by 75% of users), there is no dynamic taint propagation engine primarily because it is extremely complex to build one. The nearest competing work in this space was published in 2018 for version 57; we are now at Chromium version 117, and the current version is very different from the 2018 version. We outline the details of a multi-year effort in this paper that led to *PanoptiChrome*, which accurately tracks information flow across an arbitrary number of sources and sinks, and is to a large extent, portable across platforms.

We experimentally show that we can discover fingerprinting APIs that can uniquely identify the browser and sometimes the user, which are missed by state-of-the-art tools, owing to our comprehensive dynamic analysis methodology. For the top 20,000 most popular websites, we discover a total of 362 APIs that have the potential to be used for fingerprinting – out of these, 208 APIs were previously not reported by state-of-the-art tools.

## KEYWORDS

Javascript taint tracking, Chromium browser, fingerprinting

## 1 INTRODUCTION

As of today, JavaScript (JS) is a ubiquitous programming language which has a very rich set of APIs for creating interactive web applications, streaming media and server-side code. Specifically, the NodeJS APIs for handling file systems, asynchronous operations, DNS resolution, and threading are important building blocks of any server-side solution. Prominent examples of sites that are built on Node JS include Netflix, NASA, PayPal LinkedIn, Twitter and Medium [14]. Javascript's realm of usage is further enhanced by the Electron framework that allows web developers to create full-fledged desktop and mobile applications using a combination of JavaScript, HTML and CSS. Almost all websites and a lot of very popular desktop applications such as Microsoft Teams[8], Zoom[5], and Visual Studio Code[9] use this framework. According to a developer study conducted by StackOverflow[6], JavaScript is the most popular programming language, with over seventy percent of developers using JavaScript for development.

**Unpublished working draft. Not for distribution.**

Because of its wide usage, there are a plethora of JavaScript third-party libraries for client, server, desktop and mobile applications. Most websites contain third-party JS libraries from various domains. These libraries provide an interface for collecting user analytics (tracking and performance), session record and replay, advertisements, dynamic form management, social media sharing, shopping carts and a host of other features. All of these libraries can access the shared page state (DOM, JS variables); they have equal privileges and can alter and access data from other co-located scripts. Apart from software bugs and vulnerabilities, the security and privacy threat to the hosting site increases manifold when third and fourth-party libraries (included by third parties) get included. As per a recent survey [1], 37% of third-party scripts are known to contain undisclosed vulnerabilities – this puts all kinds of personal data such as passwords, medical data and credit card details at risk. Along with explicit information leakage, side channels and other implicit sources of information can be used to uniquely identify browsers and characterize user behavior such as using battery level indicators [33]. The entire area of *browser fingerprinting* [20] relies on such implicit sources of information that can help an adversary identify a user or browser with a reasonable degree of accuracy. Hence, to summarize, a comprehensive security analysis tool at the client side is necessary to identify JS APIs and websites that display such malicious behavior.

Taint analysis in web browsers is an established problem. Here, the flow of information is tracked from a sensitive source such as a password field to a sink, which can potentially exfiltrate the data to an unauthorized party. Analyses can either be static [27, 31] or dynamic [29, 36, 37]. A criticism of static analysis approaches is that they are either overly conservative or they miss out on capturing dynamic information, which is of vital importance in Java script. A lot of information is not available at compile time such as the contents of third-party APIs and the results of *eval* calls, where a JS statement is created dynamically. Dynamic analysis, on the other hand, is difficult to implement because it involves invasive changes to the code of the web browser and JavaScript engine (V8 in the case of Chromium). To put matters in perspective, the code size of Chromium and the V8 engine are 35 million lines and 3 million lines, respectively. Moreover, they have complex memory allocation and garbage collection (GC) mechanisms, which make it very difficult to add additional metadata to objects and track the flow of information (especially implicit flows).

Hence, many researchers [21, 25] have opted for simpler methods where they annotate JS APIs and then log their executions. This can be easily detected or it requires a complete reimplementation of the entire runtime in JavaScript. Furthermore, these methods often miss many subtle interactions and control flow based dependences. As a result, the gold standard in this area is to track flows by modifying the browser and the JS engine, which is what our nearest competing work, Mystique, did for Chrome version 57. We are currently at version 117 and in the last 60 generations a lot of fundamental changes have been made in the source code. For instance, Chromium has

transitioned from a stack-based to a register-based virtual machine, the execution pipeline has changed, and the memory management and GC systems have been completely overhauled.

We thus propose a bespoke dynamic taint analysis framework called *PanoptiChrome*, which adds roughly 7,000 lines of code to the existing V8 engine. Its novel features are as follows.

❶ It accurately captures all kinds of information flows (explicit and implicit) while supporting a variable number of sources and sinks that can be changed at runtime. ❷ It is mostly portable across Chromium versions (requires minor changes) and is platform agnostic. ❸ From a software engineering point of view, *PanoptiChrome* is fairly self-contained, where all the changes are limited to the V8 engine's Ignition module (interpreter part) only. ❹ Our solution works with V8's complex garbage collection and memory relocation framework. ❺ We test the efficacy of *PanoptiChrome* for the problem of identifying fingerprinting APIs across the top 20,000 websites. We identify 164 hitherto undiscovered APIs that are potentially fingerprinting.

Section 2 describes the background, Section 3 elaborates on the design of *PanoptiChrome*, Section 4 shows the evaluation results, Section 5 describes the related work and finally we conclude in Section 6.

## 2 BACKGROUND

In the following section, we begin with an overview of information flow analysis and its usage in various aspects of web privacy and security. We then present the reasons for adopting a static+dynamic taint analysis algorithm over static taint analysis approaches. Furthermore, we establish the need to instrument the runtime system, specifically the V8 engine.

### 2.1 Browser APIs and WebIDL

Browser APIs allow a website to access certain features such as the browser type, current date and time, hardware and software configuration, screen dimensions, etc. Since different browsers might offer different capabilities with differing syntax, the WebIDL standard formalizes the interfaces and properties that need to be offered by a compliant browser. In Chromium, these APIs are implemented as a part of the Blink rendering engine; they are exposed to web applications using the standard Web IDL specification.

Due to the variability in the nature of the devices that access a given site, along with geographical differences (detected from the IP address and time zone), browser APIs typically return different kinds of information for different devices. These values obtained from browser APIs, along with the meta information about the device allow the site to create a unique *fingerprint* for each device. The site can then use this fingerprint to track the user across different sessions and websites, even if the network used to access the site changes.

### 2.2 Information Flow Analysis

There are two kinds of information transfer channels: an explicit channel (the information is directly transferred from one object to the other via an immediate assignment) and an implicit channel (information transmission through direct and indirect control flows).

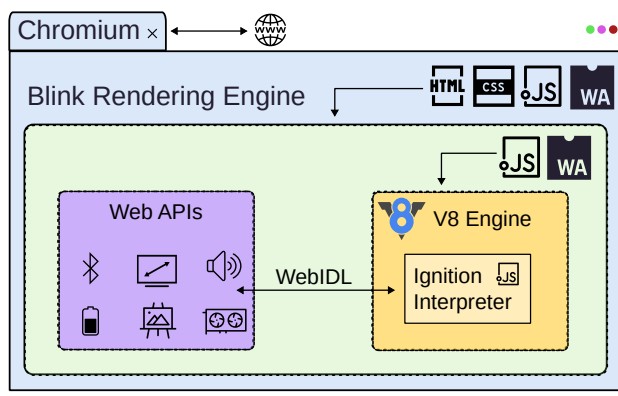

**Figure 1: Overview of the Chromium architecture**

JavaScript uses an asynchronous event-driven programming model, where functions are registered as event handlers and invoked when the corresponding event happens such as a "mouse hover" or a "mouse click". As a result, static analysis approaches aren't very effective and seldom generate accurate information/-control flow information. Another factor that complicates static analyses is the fact that JS is a deeply object oriented language where properties (variables/functions) can be added and removed dynamically from parent classes, and functions are treated as objects. They also may take a variable number of arguments. To complicate matters, JS has the *eval*[3] function that allows the interpreter to execute strings provided at runtime as code, which makes static analysis nearly impossible. Finally, the reflection API (introduced in ES6[2]) allows code to be self-modifying and have mutating properties of objects. Hence, dynamic analysis is required.

Approaches that instrument the code instead of the runtime cannot add hooks to all the objects' properties without a priori information about the properties themselves. Further, there are multiple API calls like *document.location* that a proxy object with hooks cannot wrap. Hence, all calls to the *document.location* API call cannot be intercepted. Such hooks can easily be detected by an adversary.

**Mystique** [16] is our nearest competing work that added dynamic information flow analysis (taint propagation) to the Chromium browser (in 2018). Mystique uses custom APIs that modify the underlying components of the JavaScript engine – this makes it hard to maintain it as the architecture of the web browser changes over time. Furthermore, Mystique tracks the information flow of only browser extensions and cannot track flows in the context of the hosting site or other third-party scripts. Moreover, Mystique collects variable dependencies at the granularity of functions. For explicit flows (due to assignments), an edge is created in the Data Flow Graph (DFG) from the R-value to the L-value. Implicit flows where there is data transfer from caller to callee parameters during function calls and returns are also handled in a similar manner. To handle control dependencies, all the variables in the branch path are tainted.

The limitations of Mystique are that it does not handle dynamic sources and sinks, is not designed for a register-based machine,

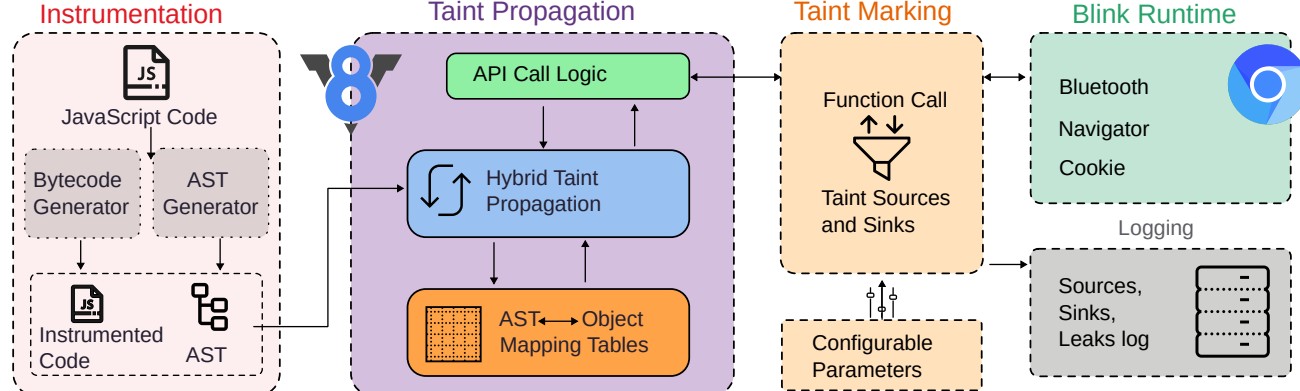

**Figure 2: Overview of *PanoptiChrome***

modifies the garbage collection and memory relocator, has extended object liveness and requires changes in the Blink rendering engine of Chromium – all of these changes reduce the portability of the design. Furthermore, because of its limited set of sinks, it misses many information leakage paths.

## 3 DESIGN OF PANOPTICHROME

### 3.1 Design Overview

To detect data leakage from the browser, *PanoptiChrome* needs to identify and mark all the values obtained from a known subset of browser APIs as *tainted*. ❶ We instrument the code generated for API methods and property (variable) accesses and add hooks (callback functions). ❷ The custom **T**aint **M**arking **E**ngine**(TME)** handles the marking of tainted values as these APIs get accessed. ❸ Once we have identified and marked the taint sources (browser APIs), the **T**aint **P**ropagation **S**ystem **(TPS)** disseminate the taint tags via explicit and implicit flows to all the objects that somehow use the tainted value (directly or indirectly). Multiple tables that are accessed using the hash of the in-memory addresses of the tainted objects are used to disseminate taint information. ❹ On invocation of an API labeled *sink*, all the parameters passed to the API are checked for their *taint* status. If we find a parameter to contain data from a tainted source, the source, sink and the parameters, we log this information to a file. The patches to Chromium's V8 engine developed for *PanoptiChrome* are available here.

### 3.2 Code Instrumentation and Data Structures

The `Bytecode Generator` in V8 walks the AST (Abstract Syntax Tree) generated by the JS parsing and analysis phase to emit intermediate code that the `Ignition` engine interprets. The `Ignition` engine in V8 is a register-based interpreter with handlers for around 230 bytecodes. 60 bytecode builders responsible for emitting the properly formatted bytecode along with 85 visitors that walk the AST, generate handlers for these 230 bytecodes. *PanoptiChrome* needs to modify only three builders and eight AST visitors to track the flow of tainted information through the execution of the JavaScript code.

In prior work, the taint marking was done when the rendering engine called a JS function. However, in our scheme we track dependences at a finer level and we can change the sources and sinks at runtime. Hence, in our case, the taint marking step needs to be intertwined the taint propagation step. For every JS object (defined in the source code), there is a runtime object (internal to V8). Whenever, we access a method or property in a JS object, we need to use the TME engine to find if we need to taint the status of the corresponding runtime object. The TME engine needs to check the list of current sources.

#### 3.2.1 Data Structures Used. *PanoptiChrome* uses multiple hash tables to store the taint status of JS objects and their corresponding runtime objects. The **Object Taint Table** (OTT) stores the "taint metadata" of the runtime object and is indexed using the `ptr` (tagged heap pointer) data member of the runtime object. This hash table stores information about all the taint sources for the given runtime object. There is an important design decision here. Should we store a list of all methods/properties via which the taint flowed to a given object's method or property? Given that prior work considers few sources, they indeed store this information. This is not a scalable solution because references to all the objects on the path will remain live, and the GC will not be able to remove them – this results in a large memory footprint.

We thus maintain two references in each OTT row: reference to the runtime object and a reference to the string encoding (runtime object) of taint sources (the runtime objects on the path are not stored). If the OTT is reachable, then all the runtime objects that it points to will also remain alive, which is something that we do not want because many objects will not have valid references to them in the original JS code left. Note that all the data structures that we add are extra ones and thus their references do not have the same degree of legitimacy as references in the JS code itself. V8 has the option of creating "weak references", which is a pointer where the destination object can be garbage collected because its reference count is not incremented due to the reference. Such weak references are used here. The **crux of the idea** here is to use two weak references: one to the runtime object and one to the string encoding of the taint sources. The *Ephemeron hash table* ensures that

if the runtime object is alive (because of references in the JS code), then the string encoding will also be alive (not garbage collected). This guarantees the existence of the string object (corresponding to the taint sources) once we reach the sink because the sink needs to be obviously alive.

The **AST Taint Table** stores the taint status of the nodes in the AST parsed for the functions in the current activation stack. *PanoptiChrome* implements **AST Taint Table** in the same fashion as Mystique. It uses a multi-level HashTable in which the first level is indexed by the frame pointer of the currently executing function, while the second level is indexed using the unique location of the node in the AST.

The link between the AST node and the corresponding runtime object is maintained using the SimpleNumberDictionary (internal to V8) that maps the location and type (VariableProxy, Property or Call) of the AST node to the corresponding runtime object. Like the AST Taint Table, the **AST to Object Map Table** contains multiple levels wherein the current frame pointer indexes the first level while the second level stores the actual mapping.

*3.2.2 Liveness of Objects.* All the tables that we add can be garbage collected, which needs to be avoided at all costs. Mystique modifies the GC itself, which is a very invasive change and harms portability and maintainability. We start with observing that all runtime objects in V8 can be referenced with the help of Handles. These are themselves not garbage collected. The Handles are stored in a HandleScope that is responsible for deallocating the Handles when the scope is destroyed. To make sure that the Handles responsible for taint tables are not deallocated, we store them in a custom PersistentHandle that is aware of the special tables and is not deallocated until the PersistentHandle is explicitly reset. The custom PersistentHandle is created at the start of the execution and is destroyed only when the execution ends. The stock V8 engine does not allow the deletion or updation of Handles added to the PersistentHandle list; hence, we introduce new interfaces that allow us to replace the unused handles with null runtime objects.

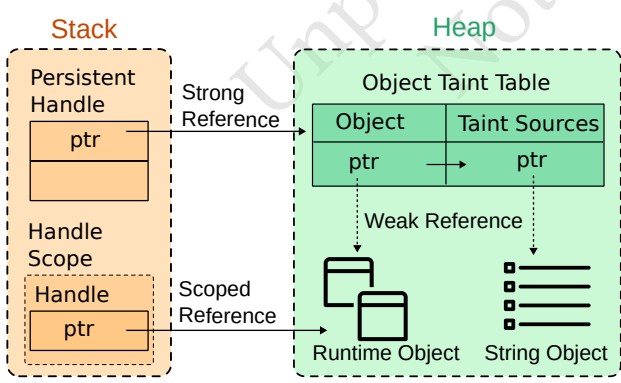

**Figure 3: Liveness in the Object Taint Table**

### 3.3 Taint Marking Engine (TME)

The role of the **T**aint **M**arking **E**ngine(**TME**) in *PanoptiChrome* is to identify the sources and sinks defined in the configuration files and taint the corresponding runtime objects. The dynamic configuration files (user-defined) contain the object's name and corresponding methods and properties to be considered as sources or sinks. *PanoptiChrome* provides the same expressiveness as OpenWPM[21] for specifying the configuration with the ability to selectively allow or deny specific properties or methods of an object. TME also filters out contexts where the sources should not be tainted. These contexts include native built-in functions (called for initial setup) and Chromium's intrinsic functionalities (like settings or a new tab). The TME receives details about the object, the API to access, values of all the parameters passed to the API and the return value. Based on the information received and the context derived from the object, TME sets the taint for the return value.

A simple lookup of the object and member name (property or method) in the custom taint configuration database is insufficient since members can be references to runtime objects. Furthermore, prototypal inheritance in JavaScript allows a child object to access all the members of superclassses. To solve these problems, TME actually needs the name of the object (similar to runtime type information in C++). The constructor's name is sufficient for this purpose. Using the constructor, the TME walks up the inheritance chain and finds the object in which the member is defined (the WebIDL specification is used to speed up this process). In some cases, when only a member is provided, the default object is *Window* (regular JS semantics). Once we find the object we check whether it is a tainted source or not. This is more elegant and generic than Mystique, which required patching all the Blink endpoints (7000+ at the time of writing this paper) and then subsequently tracking their accesses.

### 3.4 Taint Propagation System (TPS)

To ensure that the original execution is not affected while propagating taint information, *PanoptiChrome* follows a caller-saved scheme – store the original values in a set of virtual registers and allocate independent registers for storing taint metadata before starting the taint propagation routines. After the routines return, the original state is restored. To ensure proper taint propagation, we include vital information about object constructors (see Section 3.3) in the parameters that we pass to the taint routines as opposed to earlier frameworks like Mystique that did not do so. Because of their restrictive nature (fixed set of sources and sinks) and limitations imposed by the stack-based Javascript VM, this wasn't easy to do.

On function exit, the AST Taint Table and AST to Runtime Object Map are dropped (since every invocation requires a fresh AST Taint Table and map). In contrast, the Object Taint Table persists across invocations to further propagate the taint status.

Once the **TME** has marked the values obtained from a select set of browser APIs as taint sources, **TPS** sends the taint tags to other objects that receive information from the labeled "tainted" sources. *PanoptiChrome* performs an order-independent, intra-procedural analysis on the source code received by the V8 engine for execution to create the flow graph (FG) (combination of the data and control flow graphs).

Initially, an Abstract Syntax Tree(AST) is generated at the level of an individual function. Analysis at the function level is sound since even the top-level scope is considered a function(unnamed).

*PanoptiChrome* repurposes the parser and generator used by the V8 JavaScript engine to create the ASTs. Then, the generated AST is cached for future invocations. This AST is then used to construct the Flow Graph (FG) in which the vertices are nodes from the AST (56 such types in version 11.7 of V8) that represent either a property access or an API call. The FG contains directed edges between the AST nodes if there is an explicit flow of information (via the assignment operator) or an implicit flow (via conditional statements).

To reduce the overhead of taint propagation, *PanoptiChrome* does not create an FG unless at least one tainted source has been visited in the scope of the function under analysis. Once an API call has been marked tainted by the **TME**, **TPS** marks the corresponding AST node in the FG as tainted with the help of the $object \rightarrow \langle AST node \rangle$ mapping table (see Section 3.2).

Taint propagation routines are invoked only when one of the following conditions is true: ❶ an API marked as a *sink* is called, or ❷ the function returns and an object/array is created by the function. Taint propagation is then carried out by following the outgoing edges from the tainted AST node and updating the taint status of each node in the forward slice (transitive closure of nodes in the FG). Also, whenever a node is marked as tainted in the FG, the corresponding object is marked as tainted with the help of a reverse mapping table (AST node to object).

## 3.5 Logging Data

The V8 engine utilizes the concept of an `Isolate` to separate different execution contexts on the same web page. These multiple execution contexts get created due to the inclusion of numerous `iframes` in the same web page. An `iframe` is a separate webpage loaded from an entirely different origin that gets embedded in the parent website (used for advertisements, videos and analytics). For every `iframe`, a separate `Isolate` is instantiated with its individual copy of the global objects and built-in functions. Distinct isolates on the same web page execute concurrently using separate threads and behave as individual sandboxed instances of the V8 runtime. Since *PanoptiChrome* attaches all the tables required for taint marking and propagating with the `Isolate`, multiple runtime instances can execute in parallel without treading on each other's data. We log all the data for each isolate separately in a different file similar to VisibleV8 [25]; hence, the problem of inter-process synchronization is solved by design.

For every `Isolate` *PanoptiChrome* logs all the origins (multiple sub-domains can be loaded in the same `Isolate` as long as they share the same origin) along with the tainted source APIs and sink APIs invoked during the execution. Additionally, the leaks are logged (along with the string-encoded list of taint sources) whenever data from a tainted source flows into a sink API.

## 4 EVALUATION

### 4.1 Setup

We use an `AMD EPYC 7702P` powered workstation with 64 physical cores and 128 GB RAM for crawling and post-processing tasks. Log files, averaging 150KB per website, are stored on a 1 TB SSD. The crawling process utilizes the latest Chromium browser (version 117.0) compiled with our custom V8 engine (*PanoptiChrome*).

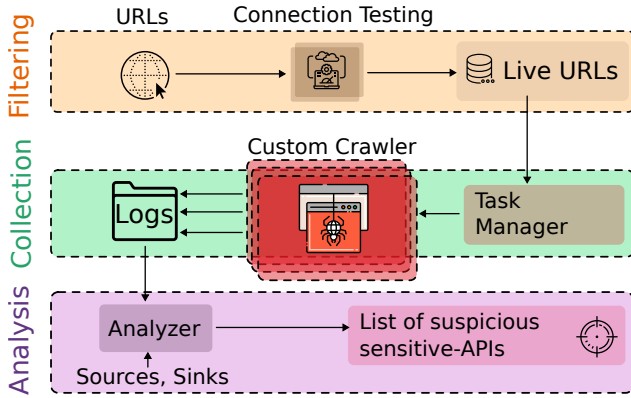

**Figure 4: Overview of the data collection pipeline**

Websites are loaded concurrently in independent browser windows with transient user profiles that get erased after each website visit. Additionally, navigation and website loading are facilitated through a commercial off-the-shelf ISP that is used by more than 38% of the active internet users in the country.

### 4.2 Crawling Methodology

As discovered by VisibleV8 [25] and OmniCrawl[15], websites frequently access various inbuilt objects and properties of the browser for various purposes such as bot detection and for optimizing their execution. Approaches that employ browser automation, like using driver frameworks such as Selenium and Chromedriver are prone to report inaccurate site statistics due to bot detection measures. To alleviate this problem, we instantiate each browser instance from the command line and pass the website URL as a parameter. Since we do not use any automation framework, our approach is virtually indistinguishable from a normal user accessing the site. The driver script to generate command line parameters and instantiate a browser is written in Python v3.7 and is responsible for closing the browser after a preset time (180 seconds).

For data collection, we use the top 20,000 websites from the Tranco list [7] as the seed URLs. We perform a connection test for each URL by requesting the HTTP header from the website. To request the HTTP header containing the website's status code, we initially attempt to connect to the website by appending HTTPS:// with the domain name from the Tranco list. If the connection succeeds, we log the schema and URL to the list of reachable URLs. If we fail to connect to the site within 15 seconds, we try with the HTTP protocol for the next 15 seconds. If we still fail, we log the URL with the corresponding error code.

**Crawling results** From our network vantage point, 61.91% of the Tranco top 20,000 websites were reachable (status code 200). Around 14% returned a 404 (not found) error, and around 24% timed out with both the HTTPS and HTTP protocols. Table 1 represents the status codes for the URLs in the list. Our crawler could log 12,846 unique origins and recorded 45,942,545 API calls and 24,486 leak entries. Furthermore, the recorded origins invoked 5,673 unique APIs, where 3,426 are DOM manipulation APIs.

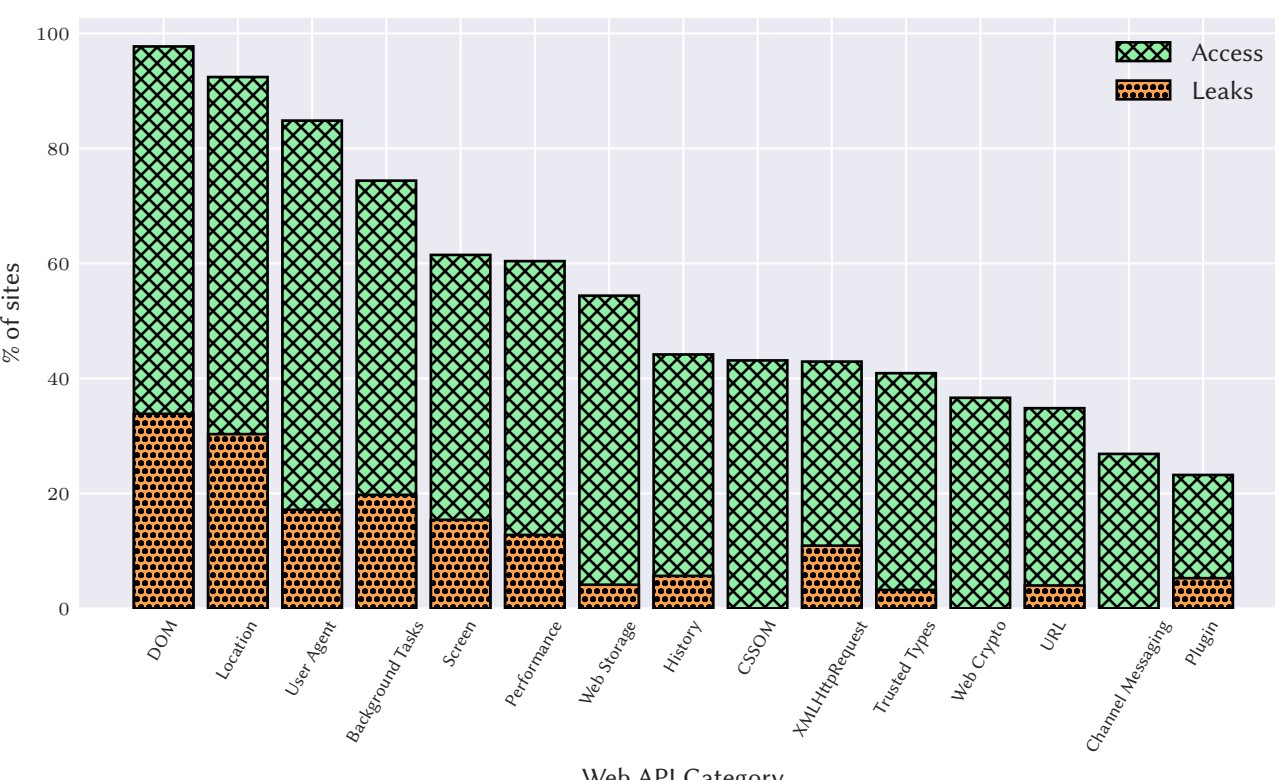

**Figure 5: API access and leak in sites vs top 15 Web API categories (based on access)**

**Table 1: Status Codes for the top 20,000 Tranco URLs**

| Status Code | Number of Sites | Percentage |
|---|---|---|
| 200-299 (Success) | 12382 | 61.91% |
| 300-399 (Redirection) | 5 | 0.03% |
| 400-499 (Client Error) | 2770 | 13.85% |
| 500-599 (Server Error) | 141 | 0.7% |
| Exception | 4701 | 23.5% |
| Total sites : | 20000 | |

## 4.3 Web API Categorization

Web API categorization is required to identify the DOM manipulation APIs that are used to get the static properties of elements in the web page. The values returned by these APIs is always same for a particular element across browsers and hence cannot be used to fingerprint the user. In our information flow analysis, we do not consider these APIs as the sources of sensitive information. To classify the APIs, the category was decided using the developer documentation provided by Mozilla Developer Network (MDN)[4]. Out of 5,673 unique API in our crawl, MDN had no categorization for 1,246 which we manually classify after analyzing the documentation. The complete list of API categories can be found here. Also, for APIs categorized in multiple categories, we give the lowest priority to the DOM category and classify the API into the higher priority bin.

## 4.4 Data Leakage from APIs

We define *data leakage* as the flow of information from any web API to any sink (storage, network). Every API invocation marks the returned data as tainted. An entry is logged whenever any tainted data reaches a sink. Out of the 5,673 unique web APIs that are accessed by 12,846 origins, our analysis reveals that data from a total of 675 unique APIs is leaked. We observe that on an average, 115 unique APIs are accessed on a website and data from 11 unique APIs is leaked. The maximum number of APIs accessed from a single origin is 531, whereas the maximum single origin leakage was found to be 144.

DOM-related APIs (such as NodeList.length) are leaked on 33.84% of the sites, followed by the Location, the Background Tasks and the User Agent category (30.31%, 19.74% and 17.15% respectively).

In the Location category the HTMLAnchorElement.hostname and the Window.location are the most commonly leaked APIs. These APIs are used to get to domain name of the page for constructing dynamic links or fetching web resources. In the Background Tasks the Window.setTimeout API is used to execute a JS function after a set amount of time. Window.navigator and Navigator.userAgent from the User Agent category are used to customize the website for different screen resolutions and sizes. The complete category wise distribution of API access and leaks for the top 15 Web API categories (based on access) is shown in Figure 5.

Out of all the APIs that get leaked on a site, 55.58% belong to the DOM category. The average distribution of API categories (in the remaining 44.42%) that get leaked per site is shown in the Figure 6.

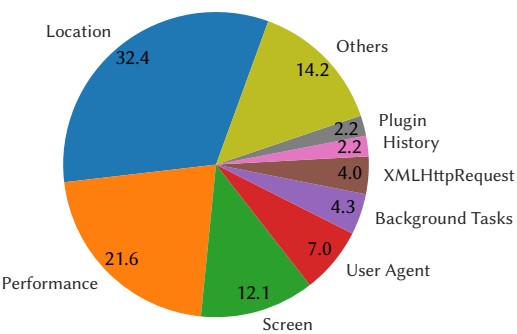

**Figure 6: Average API leak distribution per site apart from DOM APIs**

## 4.5 Fingerprinting APIs

*Sensitive APIs* are those APIs that can be potentially used for fingerprinting. *PanoptiChrome* analyzes all the parameters passed to a sink and records the sources from which the information sent over a sink is generated. Out of the 675 unique APIs that get leaked, 269 APIs are DOM related that are invoked for DOM manipulation. These are unrelated to fingerprinting because they always return the same value. 121 APIs from the remaining 406 APIs have been already classified by the previous work [35] as *sensitive* and 33 have been categorized as *URL* or *sink-related* APIs that are used for fingerprinting indirectly (as a means of ferrying already fingerprinted data).

For classification of the remaining APIs (252 APIs), we followed a method similar to prior work [35], where we investigated known fingerprinting websites for the use of the discovered APIs. In total, 78 APIs were confirmed to be fingerprinting using this method. For the remaining 186 APIs, we manually consulted the documentation for each API and the source code of the websites that use the API to establish the association with fingerprinting. 82 APIs were manually classified to have the potential to be used for fingerprinting, while 48 APIs were classified as sinks or providing URL-related data. **To summarize, we discover** a total of 362 APIs (121 + 33 + 48 + 78 + 82) that are probably being used for fingerprinting or have the potential to be misused. The complete list of APIs can be accessed here. Out of these 362 APIs, 208 APIs were previously unreported by state-of-the-art works.

*4.5.1 Effect of Co-location of API Calls on Precision and Recall.* Out of the 675 APIs that are leaked by the sites, 39.85% are used for DOM manipulation and 89.16% of the remaining APIs (total minus DOM) were confirmed to have the potential to be used as fingerprinting vectors (sensitive). For each sink, *PanoptiChrome* reports a list of sources that are used to compute the tainted value. For each such list, we check if it contains more APIs than a pre-defined

threshold. For this experiment we do the following: if this threshold is breached, we mark all the APIs in the list as *sensitive*. This follows from the observation in [12, 24] that browser fingerprinting often clubs data from multiple sources together to form a unique identifier. After removing DOM manipulation APIs, we vary the threshold from 0 (all sources included) to 12 (maximum number of seeds found in a single leak) and plot the resulting distribution of percentage of suspicious APIs found in Figure 7. It can be observed that the percentage of APIs detected to be sensitive increases with the threshold (precision increases at the cost of recall). With a threshold of 0, 406 APIs are marked for further analysis (with 89.16% being sensitive), while with a threshold of 10, only 92 APIs are marked for further analysis. Out of the 92 APIs marked, 96.74% are confirmed to be sensitive.

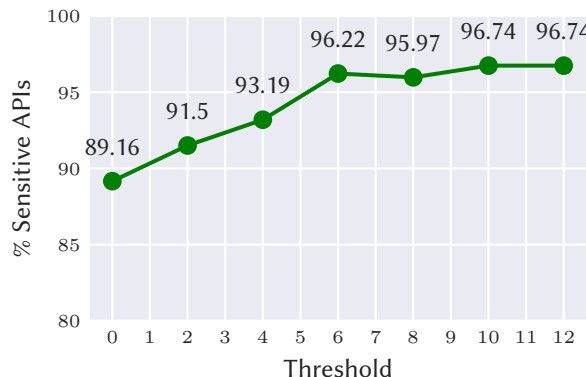

**Figure 7: Relationship of the #sensitive APIs found with the threshold value**

**Comparison with State-of-the-art:**

For comparative analysis with the state-of-the-art tool for fingeprinting – BFAD[35] – we collected the API logs using VisibleV8 [25] for the Tranco top 1000 sites. The results are shown in Figure 8. Only 608 of these 1000 sites were reachable from our network vantage point. BFAD confirms a total of 68 APIs that can potentially be used for fingerprinting (are sensitive). *PanoptiChrome* discovers a total of 438 unique APIs from which the data is leaked. Out of these 438 APIs, 183 APIs are used for DOM manipulation, hence are not considered for fingerprinting. In the remaining 255 APIs, we verify 237 APIs to be a potential vector for fingerprinting either manually or by using the known fingperinting approaches.

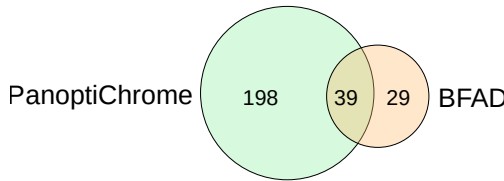

**Figure 8: Comparison with the state-of-the-art, BFAD[35]**

**Table 2: Summary of information flow analysis in different JavaScript runtimes. FGTT stands for fine-grained taint tracking**

| Work | FGTT | Implicit flows | Custom sources | Completeness | Upgradability | Platform agnostic |
|---|---|---|---|---|---|---|
| Vogt et al.[37] | ✓ | ✓ | ✗ | ✗ | ✗ | ✗ |
| DOMsday[32] | ✓ | ✗ | ✗ | ✗ | ✗ | ✗ |
| WebPol[10] | ✓ | ✓ | ✓ | ✓ | ✗ | ✓ |
| Runtime monitoring[13] | ✗ | ✗ | ✗ | ✗ | ✓ | ✓ |
| Crowdflow[28] | ✓ | ✗ | ✗ | ✓ | ✗ | ✗ |
| FPDetective[11] | ✗ | ✗ | ✗ | ✗ | ✓ | ✓ |
| JSgraph[30] | ✗ | ✗ | ✗ | ✗ | ✓ | ✓ |
| Visible V8[25] | ✗ | ✗ | ✗ | ✗ | ✓ | ✓ |
| Mystique[16] | ✓ | ✓ | ✗ | ✓ | ✗ | ✗ |
| 25 million flows[29] | ✓ | ✗ | ✗ | ✗ | ✗ | ✗ |
| *PanoptiChrome* | ✓ | ✓ | ✓ | ✓ | ✓ | ✓ |

## 5 RELATED WORK

Dennings [18, 19] pioneered the formalization of static analysis approaches in 1970s. Fenton [22] then studied purely dynamic monitors for managing information flows. Much of the later work has focused on adapting the work of Denning and Fenton to different languages and proposing solutions with various limitations.

Dynamic analysis techniques using virtual machines [23], source code instrumentation [34], and runtime instrumentation [13] have been employed for numerous use cases ranging from JS execution visualization [30] and record/replay [34] to privacy and security analysis of browser extensions [16] and policy enforcement[10]. Table 2 summarizes the features and limitations of existing information flow approaches for JavaScript engines.

### 5.1 Augmenting Browsers with Taint Tracking Capabilities

Vogt et al.[37] supplement dynamic taint tracking with static analysis to detect DOM-XSS vulnerabilities. They use static analysis to propagate taint information along implicit flows. On the same lines, in reference [29], the authors instrument the Chromium browser to track tainted strings (limited use case). Their goal was to detect and validate DOM-XSS vulnerabilities. Based on the source and context of the tainted data, they automatically generate the breakout sequence to validate the vulnerability. Like [29], Domsday [32] also instruments the Chromium browser to detect DOM-XSS vulnerabilities. The authors add one byte to each string object to keep track of the encoding and decoding functions and also the provenance of data. This is also a limited use case. Another such work is FP-Detective [11] that only looks at font-related APIs.

Bauer et al. [13] treat the V8 JavaScript engine as a black box and track information flow only across the Blink-V8 boundary; this can be used to sandbox scripts based on their respective origins. Their coarse-grained information flow approach cannot handle implicit flows and cannot reason if a source API is exploited for illegitimate use. In Crowdflow [28], the authors aim to minimize the limitations of information flow tracking by probabilistically switching between partial taint tracking and information flow monitoring in a distributed setting. The clients report a violation to an aggregator that takes appropriate action. Similar to Domsday [32], CrowdFlow

employs heavyweight instrumentation and uses fixed set of sources and sinks that are tailored to detect XSS-based vulnerabilities.

*PanoptiChrome* is much more generic than all the prior work and is not meant to target taint information for any specific kind of data types (or object types). Its taint tracking is also much more fine-grained. Unlike prior work, it does not rely on any custom JS engine that only handles a subset of the language; it can handle any site that the Chromium browser can handle.

### 5.2 Study of Third-Party Data Exfiltration

In this space, the closest approaches that are similar to *PanoptiChrome* are Mystique[16], Jest [17], Ichnaea [26], and JSFlow [23]. JSFlow [30] uses a bespoke JS interpreter for a subset of JavaScript. Jest [17] is a source-code instrumentation-based approach that converts every statement and expression to a function call, and these instrumented functions are responsible for implementing the dynamic analysis methods. Ichnaea [26] is built on top of Jalangi [34], which is also a source-code instrumentation-based approach. The instrumentations proposed by both Jest and Ichnaea can be detected easily by an adversary [15].

## 6 CONCLUSION

We showed in this paper that it is indeed possible to build a comprehensive dynamic taint tracking engine that is completely generic and is portable across platforms and browser versions to a large extent. This was achieved by limiting the changes to a small part (interpreter) of the V8 engine and suggesting ingenious solutions for dynamic addition of sources/sinks, creating persistent handles to circumvent the issues caused by the GC and memory relocation engines and optimizing the process of taint propagation by using an on-demand algorithm. We used *PanoptiChrome* to perform a detailed characterization of the information leakage in the top 20,000 websites. We further use the locality information inherent in the logs generated by *PanoptiChrome* to significantly reduce the set of APIs to be considered for manual analysis. The need for having *PanoptiChrome* is attested by the fact that we discover 208 APIs that were not known to have a fingerprinting character.

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
