# OpenReview forum: "PanoptiChrome: A Modern In-browser Taint Analysis Framework"
_ACM.org/TheWebConf/2024/Conference — TheWebConf24 Oral_

### Official Review · Reviewer_Nn8Q · 2023-11-19

**Novelty:** 4
**Technical Quality:** 6

**Review:**

This work presents a new taint engine built atop Chrome's V8, which aims at providing a generic framework to enable taint analyses with a maintainable and configurable way, both of which are hard issues that have been long-standing problems to the community. I do believe that a generic engine like the proposed one can have massive impact on the community, as future work might no longer be tasked with introducing their own taint analyses, or building atop the non-maintainable patches that happen once every 5 years [29] and then [32] from which the taint engines were then used in subsequent research.

Unfortunately, the engine's taint propagation engine is simple and does not keep track of the way that taint propagates through the execution, such that more complex techniques such as [29] and [32] cannot be replicated. Nonetheless, the taint engine is a strong technical contribution.

Based on the descriptions of the work it is hard to understand that the claims of easy maintainability of the patches is substantiated. I would not expect the majority of readers to be knowledgable of the intrinsics of V8, thus, providing some insights into how patches would need to be maintained is crucial to substantiate the claims. E.g., does maintenance only mean we have to keep track of new byte code instructions and new AST nodes? How would this change if we want to make the taint propagation tracking more sophisticated, do we need further instrumentation on the bytecode level?

The authors use the engine to study previously unknown "potential fingerprinting vectors". Which is the main shortcoming of this work imo. In particular section 4.4 performs a simple lookup on the invoked APIs and the which ones of those end up in a sink that is either storage or network related. The authors label any such flow as a data leak which inflates the severity of those flows significantly. First, putting something into local storage has no direct ramification for security or privacy. Second, I am not convinced that the source classes as defined right now paint a good picture of sensitive sources. I.e., the data flows from HTMLAnchor elements or Trusted types seem to me very much intended functionality and not leaking behavior. Similarly, the user agent will just be sent with any network request anyway (unless we consider a proxy that changes it out in the middle). So those can at best be marked data flows and not leaks.

When classifying the unknown fingerprinting APIs, it is my understanding that you visited fingerprinting websites and just labelled all APIs used as fingerprinting related. In contrast [35] performs an analysis based on proximity of the invoked function calls, which provides a much stronger signal that those APIs are actually part of the fingerprinting routine. AFAIU, the APIs you label as fingerprinting could be used in a completely different part of the application and have legitimate use cases. E.g., looking at the list of `potential_fingerprinting_apis.md` there are various ones that seem a bit off, e.g., `BatteryManager.level`, `Crypto.getRandomValues`, `IDBTransaction.objectStore`, `MimeType.*`, `TrustedTypePolicy.*`, ...

Pros:
- Strong technical contribution of a supposedly maintainable taint engine
- Making the taint engine available to the community

Cons:
- Main claims of the paper surrounding fingerprinting are shallow
- Evaluation/Description of technical claim of maintainability is missing (perception could be changed over discussion)

Nitpicks:
- you use "Java script" instead of "JavaScript" in some places
- try to stay more objective in the use of language (e.g., ingenious)

**Questions:**

- Could the TPS be augmented to handle more complex taint propagation tracking to support [29] and [32]? If so, would the patches remain as "upgradeable"? Based on the descriptions I would assume so.
- Does the decreased performance of BFAD compared to the original work come from the different dataset, i.e., top 1k vs top 10k and different top lists?
- Why does PanoptiChrome not find a super set of the things that BFAD finds?

**Reviewer Confidence:**

4: The reviewer is certain that the evaluation is correct and very familiar with the relevant literature

**Scope:**

4: The work is relevant to the Web and to the track, and is of broad interest to the community

---

### Official Review · Reviewer_5QBG · 2023-11-21

**Novelty:** 4
**Technical Quality:** 3

**Review:**

Pros:

- The Design explanations for PanoptiChrome and decisions during the implementations are well motivated and explained and easy to follow.

- I very much appreciate that several diagrams explain the internals of the tool and the structure are included, and that they make it easy to understand what was done where and why. Also, I like that many of the numbers that are important for the results are presented nicely in the different figures.

Cons:

- Some claims should be toned down, e.g., "our approach is virtually indistinguishable from a normal user accessing the site" (l. 557) as there are multiple things that one can use to still distinguish this, for example via mouse movements, user interactions, rendering time, etc. Better say that you tried to make the approach not easily detectable as an automated crawl.

- Please add citations for some claims, e.g., "Most websites contain third-party JS libraries from various domains" (l. 65) e.g. [3].

- In some parts, the paper lacks some details:
--> Are only main frame invocations counted as API calls or also those from third-party iframes? If iframes are also logged (which they should from a technical point-of-view), how many of the 12,846 unique origins are still part of the initial tranco set? If they are not counted, why is the number of reached sites lower than the number of origins with detected API calls? This essential part of the methodology should be clarified such that the reader can easily see the bigger picture of the results.
--> In the comparison with BFAD there are 29 cases that BFAD detected but Panoptichrome was not able to detect. You should add an explanation for those APIs to the paper: Which are those APIs? Why did Panoptichroe miss them but BFAD not?
--> Minor thing: In 2.2 the eval function is mentioned, however, also other JS APIs can do string-to-code conversion (e.g. setInterval / setTimeout).

- It would be a nice addition to have a comparison of the found fingerprinting APIs to known Web browser APIs that are leaking (e.g., https://privacytests.org/). Also presenting a few or at least one of the newly discovered APIs in a case study would be nice to make it easier for the reader to classify the results and their impact.

- The paper only compares the tool with other (partially old) chrome-based tools and names Mystique from 2018 as the "nearest competing work". However, there are also frequently maintained taint tracking versions of Firefox (e.g. [1], using FF 116, maintained since 2021, last commit August 2023) which are also used in scientific publications e.g. [2]. The paper would benefit from a comparison between Panoptichrome and Foxhound such that future work knows which one is better suitable for the purpose of tainting flows. Notably, I do not know how Foxhound works exactly and if it can do what Panoptichrome is doing, still I think that the paper should at least mention the Project as related and explain the differences.

- The results of the crawling for the evaluation are far behind the results of related literature that used the tranco list in terms of success rate. You report a success rate of 61% (61.91% for measurement, 60.8% for BFAD comparison) while the success rate of other crawls in literature that used tranco is much higher, e.g. 83.89% success rate in [3] or 81.74% in [4]. The paper should include an explanation and comparison with other works on crawling here as it seems that there was something wrong in the crawling process, most probably network issues due to the high number of exceptions/timeouts. In addition to that the methodology has the limitation that connecting to the Websites via an unencrypted HTTP connection might influence the soundness of the results as anybody on the connection could inject arbitrary scripts into the response, which should be mentioned as a limitation.

- Minor issues:
  - The figures 1-4 are never mentioned/referenced in the text
  - Missing "," between PayPal & LinkedIn (l. 41)
  - "Java script" -> "Javascript" (l. 93)
  - Mystique (l. 112) is missing a citation


[1] Project "Foxhound" - https://github.com/SAP/project-foxhound

[2] Klein et al. - Hand Sanitizers in the Wild: A Large-scale Study of Custom JavaScript Sanitizer Functions

[3] Steffens et al. - Who’s Hosting the Block Party? Studying Third-Party Blockage of CSP and SRI

[4] Roth et al. The Security Lottery: Measuring Client-Side Web Security Inconsistencies

**Questions:**

- Please explain the differences in susccess rates of your crawls and other crawls in literature.
- Please explain the confusion about iframes and the number of origins with API calls (see Review).
- Regarding the 29 cases that BFAD detected but Panoptichrome not: Which are those APIs? Why did Panoptichroe miss them but BFAD not?
- Why did you not even mentioned Project Foxhound in the paper?

**Ethics Review Description:**

No issues

**Reviewer Confidence:**

3: The reviewer is confident but not certain that the evaluation is correct

**Scope:**

3: The work is somewhat relevant to the Web and to the track, and is of narrow interest to a sub-community

---

### Official Review · Reviewer_6NB2 · 2023-11-23

**Novelty:** 3
**Technical Quality:** 6

**Review:**

Pros:
- The work has a significant technical artifact capable of implementing taint tracking in a customizable way.
- Taint tracking implementation should be useful to the community.
- Authors demonstrate that the taint engine is capable of identifying interesting flows by collecting privacy-relevant flows in the wild.

Cons:
- Unclear how customizable the taint engine is to specific problems.
- If the focus of the paper is the taint tracking engine, it would be useful to demonstrate more use cases than just the one from the paper.

One of the drawbacks to making a general purpose taint engine is that specific problems require that the taint engine make different tradeoffs regarding precision and under tainting and over tainting. In a general purpose engine, it can be harder for users to get what they want out of an engine. For example, if a tainted string is concatenated with a non-tainted string, and then the result is substringed such that it no longer contains any data from the tainted string, should the result be tainted? In some use-cases, the answer might be yes and no in others. In addition, using the IDL to intercept and make the specification of sources and sinks is definitely useful, but how does the system handle tracking functions that are provided by V8 outside of the IDL definitions? It would help to demonstrate for more use-cases than just the one from the paper if this taint engine is useful for general purpose, like the article is claiming.

**Questions:**

Question:
- Does using the IDL limit you to only environment provided functions? Can the tool track flows inside of v8 native functions?
- What control do users have over the propagation rules? For some uses, you may want very precise byte-level tracking, for others more coarse-grained tracking is better?

**Reviewer Confidence:**

3: The reviewer is confident but not certain that the evaluation is correct

**Scope:**

4: The work is relevant to the Web and to the track, and is of broad interest to the community

---

### Official Review · Reviewer_B2Cj · 2023-11-23

**Novelty:** 4
**Technical Quality:** 6

**Review:**

Description

The paper presents an information-flow tracking for a very recent version of
the JavaScript engine, V8, which is used by Google Chrome. Compared to other
similar systems, PanoptiChrome instruments the JS engine's code and not the
entire web browser. This allows for much better information-flow tracking,
compared to the inaccuracies that can be introduced when tracking is happening
at the browser level.

- Important engineering framework that can assist in further research carried
  out by the community for matters of web application analysis (at the client
  side of the JavaScript code).

- The authors list anonymously the patches for the V8 engine, and I believe
  this is a clear intension of the authors for open-sourcing their framework
  and releasing it to the rest of their community.

- Unclear verification of data leakage.

- It would be good to see if this effort is very custom to V8 and if we can
  extract some highlights that can potentially generalize this approach to
  other JavaScript engines.

- It would be good to have more applications based on the proposed framework
  demonstrated in the paper.

Details

Framework. The engineering effort behind PanoptiChrome is substantial, and the
result looks useful for the research community.

Data leakage. In Section 4.4, you use PanoptiChrome to discover data leakage of
JavaScript APIs that are used out in the wild. I wonder, how can you verify
that all leaks are correctly reported and here are no bugs in your framework
that can lead to over-reporting (e.g., due to over-tainting, perhaps)?

Generalization of Techniques/Other applications. PanoptiChrome is a
modification of V8, which is very popular and this is fine. I was wondering if
parts of the methodology could stand as generalized concepts that can be
applied to other JavaScript engines, as well. I am also wondering, if V8 is
friendly in receiving such modifications compared to other engines. Moreover,
it would be a good addition if authors could sketch the steps of applying
PanoptiChrome for realizing other applications, apart from the ones they have
in Section 4.

**Questions:**

- How can you verify that APIs that leak information are not false positives
  due to over-tainting?

- Is it possible that some concepts of your approach are applicable to other JS
  engines?

- Could you demonstrate how PanoptiChrome can stand as a framework for building
  other applications?

- Are you going to release PanoptiChrome?

**Ethics Review Description:**

Nope

**Reviewer Confidence:**

4: The reviewer is certain that the evaluation is correct and very familiar with the relevant literature

**Scope:**

4: The work is relevant to the Web and to the track, and is of broad interest to the community

---

### Official Review · Reviewer_mmTW · 2023-11-24

**Novelty:** 4
**Technical Quality:** 4

**Review:**

# Summary
This paper presented a new dynamic taint tracking implementation in the recent version of the Chromium browser. The design is claimed to be portable across Chromium versions and requires limited modifications in the V8 engine's Ignition module. The authors evaluated the effectiveness with 20,000 real-world websites and detected some new APIs that can potentially be used for fingerprinting.

# Strengths
- This paper proposed a taint analysis framework to support analysis for custom-defined sources and sinks.

- The design requires limited modifications in only one module and is thus easily portable.

- The authors evaluated the framework using many real-world websites.

- The source code and experiment data are available. This is a great contribution to the community.


# Weaknesses
- The motivation and technical contribution of another implementation of dynamic taint tracking system in the browser are weak.

- The security benefit of the system is not sufficiently demonstrated.

- There is no evaluation on the performance of the system. This is critical as high performance overhead is the main limitation of dynamic IFC systems.

- Some technical details are poorly explained, and some illustrations are hard to follow.

- The comparison with prior works is insufficient and might be biased.

- Some claims are not clearly supported.


# Other comments

## Motivation and technical contribution
It is true that Mystique was implemented for an old version of Chromium and it mainly was used to track information flow of browser extensions. These do not sufficiently justify the novelty and technical contribution of PanoptiChrome. How hard would it be to adapt Mystique's design to newer version of Chromium except for engineering efforts? How hard would it be to enable taint tracking for web scripts in Mystique? Aren't both the extension code and web scripts executed in V8 but just in different isolates?

Mystique could propagate taints to the HTML code and also the browser storage. How does PanoptiChrome support those?

It is motivated that the current Chromium uses a register-based VM. I do not identify what specific new corresponding challenges the authors have addressed in implementing PanoptiChrome.

The research is also not well motivated in the introduction. It discussed first that JS is widely used, including in the server-side and in desktop application development, which are not relevant to this work. It then discussed some privacy threat caused by over-privileged JS code. But then the paper focuses only on fingerprinting, which is too narrow.

## Unclear technical details
Section 3.1 is unclear. Is the instrumentation for all APIs or just those defined in the configuration?

The authors mentioned "explicit and implicit" and "direct and indirect" many times. Can the authors provide some examples to explain these conditions better? For instance, in Line 172, what is the implicit channel of information transfer, and what are the direct and indirect control flows?

What is the priority order of other categories except for DOM in section 4.3?

## Comparison
The authors mentioned "Mystique is our nearest competing work". Why did not they compare it with their work?

Table 2 shows that WebPol is close to this work except for upgradability. Why did not the authors compare with it?

The characterization in Table 2 does not look convincing. How do the authors decide FGTT, Upgradability, and Platform agnostic? Clearly illustrating the criteria may strengthen it.

## Limited demonstrated security benefit
The discovery of some potential fingerprinting APIs is not a major contribution to the community. The users and developers cannot do much even knowing about them. The authors could have applied their techniques to detect much more severe threats on the Web, such as XSS, credential exfiltration, etc., especially those that cannot be found by existing techniques. This would strengthen this work greatly.

## Questionable claims/statements
How do the authors prove that PanoptiChrome captures all kinds of information flows? Do the authors have any evidence to support such a strong claim? This claim is not convincing to me because the implementation of PanoptiChrome is limited to the V8 engine. Did the authors consider some specific scenarios, such as storing a tainted value in a DOM element and accessing the value using DOM manipulation APIs? Furthermore, can this solution handle partially propagated cases, such as when a tainted string is sliced and passed to another object?

It is written in L45-47 that "Almost all websites ... use this framework", which is clearly false.


## Other minor issues
In the experiment, PanoptiChrome only visited the websites without any operation. Some taint propagation may incur only if the users interact with the websites. Simply visiting the websites may not identify sufficient fingerprinting APIs.


# Presentation
 - Section 3: Please ensure consistent terminology usage. For instance, "AST to Object Map Table" (Line 365), "AST to Runtime Object Map" (Line 450), and "AST <-> Object Mapping Tables" (Figure 2).
 - Figure 2, 3, 4: Please consider using links when referencing those figures.
 - Line 41: "Node JS"; "PayPal LinkedIn" missing comma.
 - Line 93: "Java Script"

# Post-rebuttal update

Thanks the authors for their response! The authors have provided explanation on the difference with Mystique, which indeed is quite difficult to port to newer versions of the Chrome browser. I am very surprised that the authors added support of taint propagation to the DOM and browser storage in just a few days. I am also glad to learn that the authors can demonstrate other security use cases of the system, although there is no sufficient supporting evidence. I have updated my review.

**Questions:**

- Could your method be implemented in other browsers (e.g., those not using V8) with limited changes? How difficult would it be?

- Why did not you compare with the related work?

- Can you clarify some of the claims that look exaggerated in the paper?

**Reviewer Confidence:**

4: The reviewer is certain that the evaluation is correct and very familiar with the relevant literature

**Scope:**

4: The work is relevant to the Web and to the track, and is of broad interest to the community

---

### Decision · Program_Chairs · 2024-01-22

**Decision:**

Accept (Oral)

**Comment:**

This paper presents a dynamic taint tracking mechanism that targets modern Chromium browsers. One of the main goals of the proposed technique is to be easily maintainable and portable across different versions, as prior taint tracking frameworks have not been maintained through time. The authors leverage their framework to conduct a study on the top 20K websites, and discover a wide range of APIs that can be used for fingerprinting. Overall, the reviewers agreed that the presented system required significant engineering effort, and by open-sourcing it the authors are providing a valuable tool to the research community. The reviewers also appreciated the study on detecting potential fingerprinting APIs. At the same time, the reviewers also voiced certain concerns about this submission. This included the limited demonstration of the system's security benefits, the need for additional experimental evaluation (especially in regards to the system's performance), and the lack of a comprehensive comparison to prior work.

 ---